# A Mixture of Essential Oils from Three Cretan Aromatic Plants Inhibits SARS-CoV-2 Proliferation: A Proof-of-Concept Intervention Study in Ambulatory Patients

**DOI:** 10.3390/diseases11030105

**Published:** 2023-08-09

**Authors:** Christos Lionis, Elena Petelos, Manolis Linardakis, Athanasios Diamantakis, Emmanouil Symvoulakis, Maria-Nefeli Karkana, Marilena Kampa, Stergios A. Pirintsos, George Sourvinos, Elias Castanas

**Affiliations:** 1Clinic of Social and Family Medicine, School of Medicine, University of Crete, 71003 Heraklion, Greece; elena.petelos@med.uoc.gr (E.P.); linman@med.uoc.gr (M.L.); a.diamantakis@yahoo.gr (A.D.); esymvoulakis@uoc.gr (E.S.); mnkarkana@gmail.com (M.-N.K.); 2Department of Health, Medicine and Care, General Practice, Linköping University, SE-581 85 Linköping, Sweden; 3Department of Health Services Research, CAPHRI-Care and Public Health Research Institute, Maastricht University, 6229 HX Maastricht, The Netherlands; 4Laboratory of Experimental Endocrinology, School of Medicine, University of Crete, 71003 Heraklion, Greece; kampam@uoc.gr (M.K.); castanas@uoc.gr (E.C.); 5Department of Biology, School of Sciences and Technology, University of Crete, 71003 Heraklion, Greece; pirintsos@uoc.gr; 6Botanical Garden, University of Crete, 71003 Rethymnon, Greece; 7Laboratory of Clinical Virology, School of Medicine, University of Crete, 71003 Heraklion, Greece; sourving@uoc.gr

**Keywords:** SARS-CoV-2, COVID-19, natural products, essential oils, effectiveness

## Abstract

Introduction: The need for effective therapeutic regimens for non-critically ill patients during the COVID-19 pandemic remained largely unmet. Previous work has shown that a combination of three aromatic plants’ essential oils (CAPeo) (*Thymbra capitata* (L.) Cav., *Origanum dictamnus* L., *Salvia fruticose* Mill.) has remarkable in vitro antiviral activity. Given its properties, it was urgent to explore its potential in treating mild COVID-19 patients in primary care settings. Methods: A total of 69 adult patients were included in a clinical proof-of-concept (PoC) intervention study. Family physicians implemented the observational study in two arms (intervention group and control group) during three study periods (IG_2020, n=13_, IG_2021/22, n=25_, and CG_2021/22, n=31_). The SARS-CoV-2 infection was confirmed by real-time PCR. The CAPeo mixture was administered daily for 14 days per os in the intervention group, while the control group received usual care. Results: The PoC study found that the number and frequency of general symptoms, including general fatigue, weakness, fever, and myalgia, decreased following CAPeo administration. By Day 7, the average presence (number) of symptoms decreased in comparison with Day 1 in IG (4.7 to 1.4) as well as in CG (4.0 to 3.1), representing a significant decrease in the cumulative presence in IC (−3.3 vs. −0.9, *p* < 0.001; η^2^ = 0.20) on Day 7 and on Day 14 (−4.2 vs. −2.9, *p* = 0.027; η^2^ = 0.08). Discussion/Conclusions: Our findings suggest that CAPeo possesses potent antiviral activity against SARS-CoV-2 in addition tο its effect against influenza A and B and human rhinovirus HRV14 strains. The early and effective impact on alleviating key symptoms of COVID-19 may suggest this mixture can act as a complementary natural agent for patients with mild COVID-19.

## 1. Introduction

Since the coronavirus disease (COVID-19) pandemic outbreak in 2019, an unprecedented international effort has been undertaken to identify the underlying cause (the severe acute respiratory syndrome coronavirus 2 (SARS-CoV-2)) and for the detailed analysis of its genome [1]. A global effort has also been directed towards identifying effective therapeutic regiments [2,3,4] and developing effective vaccines [5,6,7,8,9]. Several established pharmaceutical molecules and patients’ plasma have been tested as drug candidates for treating COVID-19, with heterogeneous results (see [10,11,12] and references therein). Among the many products tested against COVID-19 [13,14], several natural products, including herbal extracts [15,16], have also been assayed (critically reviewed in [17] and references therein), mainly targeting the viral proteases.

Much interest has been shown regarding the role of primary health care (PHC) in pandemic times; evidence highlights distinct challenges to integrating and supporting PHC to mount an effective response against infectious disease epidemics [18]. However, the contribution of PHC in terms of research relevant to the identification and testing of therapeutics against COVID-19 has been limited in Europe.

Our group has conducted a large number of studies on the use of aromatic plants as an effective treatment of common health problems. A combination of three aromatic plants’ essential oils (CAPeo) (*Thymbra capitata* (L.) Cav., *Origanum dictamnus* L., *Salvia fruticosa* Mill. Ref. [19] and references therein) has demonstrated effectiveness against upper respiratory tract viral infections in humans [20,21]. Plant material has been identified by one of the authors (SP) and voucher specimens of the three species have been deposited at the Herbarium TAU of the Aristotle University of Thessaloniki (UOCSP101-1, UOCSP101-2 and UOCSP101-3). The in vitro studies revealed the efficacy of CAPeo against influenza A and B and human rhinovirus HRV14 through the inhibition of the nuclear translocation of viral nucleoproteins [22], resulting in impairment of the viral protein transcription. In addition, the safety of CAPeo has also been demonstrated in humans (administered in soft gels, 1 mL/day of a 1.5% essential oil combination in extra virgin olive oil) [21]. Furthermore, our preparation was tested in vitro against SARS-CoV-2-infected cells. Our findings indicate it can promote the survival of cells following infection, thus reducing viral replication both following pre- or co-incubation with CAPeo [23].

Based on the above-published experiences, it was considered essential to design and implement a proof-of-concept (PoC) intervention study (early-stage evidence that something works as it has been intended) with a control group of patients with mild COVID-19 attending PHC services on the island of Crete in Greece. Our primary objective was to explore to what extent the CAPeo can significantly alleviate the general and local symptoms of the disease. In parallel, the efficiency of CAPeo mixture was examined.

## 2. Materials and Methods

### 2.1. Setting

The private primary healthcare center with 3 GP practices is located in a town near Heraklion, Crete’s capital. The study was conducted from August to October 2020 and from August 2021 to February 2022, periods during which different strains of SARS-CoV-2 were predominant. During the first period of data collection, the Alpha (B.1.1.7, VOC: 18-Dec-2020 with earliest documented samples in the United Kingdom, September 2020) and Beta (B.1.351, VOC: 18-Dec-2020 with earliest documented samples in South Africa, May 2020) strains were predominant, while during the second period of data collection, the Delta (B.1.617.2, VOI: 4-Apr-2021, VOC: 11-May-202) and Omicron (B.1.1.529, VUM: 24-Nov-2021, VOC: 26-Nov-2021) strains were predominant [24,25].

### 2.2. Participants

Adult patients reporting to a single primary care unit with symptoms related to an upper respiratory tract infection and suggestive of a SARS-CoV-2 infection were invited to participate in the study after the SARS-CoV-2 infection was confirmed by a real-time PCR (rtPCR) test performed in the regional COVID-19 reference centre (the Laboratory of Clinical Virology of the University of Crete). Eligible participants were (randomly) allocated to the intervention group (IG) and the control group (CG). Thirteen patients from the first period of recruitment all participated in the IG (n = 13; Aug-Oct 2020) and twenty-seven in the second (n = 25; Aug 2021-Feb 2022) with the CG (n = 31; Aug 2021-Feb 2022) (Figure 1, Table 1, Appendix A). In addition, information regarding demographics, medical history, smoking habits, symptoms, and signs was recorded in a pre-tested questionnaire for all study participants.

### 2.3. CAPeo Production and Phytochemical Analysis

Spanish oregano (*Coridothymus capitatus* (L.) Rchb. F. synonym of *Thymbra capitata* (L.) Cav.), Dictamnus or Cretan dittany (*Origanum dictamnus* L.), and Greek sage (*Salvia fruticosa* Mill.) were cultivated under total good agricultural practice and high precision agriculture (based on an ecological niche modeling tool we have recently developed [26]) to maximize their essential oil composition and content. A barcoding monitor of each batch was always performed to ensure genotyping accuracy. The essential oils of the three plants were produced from collected leaves of the plant material and air-dried in a dark at room temperature (25 °C) for ten days. Essential oils were extracted by steam distillation of the dried plant leaves under GMP conditions. The final extract contained four parts of *Corydothymus capitatus* (L.) extract, two parts of *Salvia fruticosa* (Mill.) extract, and one-part of *Origanum dictamnus* (L.) extract.

For analysis, after steam distillation, 1 mL of volatile oils was diluted with 2 mL of ether, filtered through anhydrous sodium sulfate to remove water traces, and stored at 4 °C. Analysis was performed, as described previously [21], by gas chromatography–mass spectroscopy (GC-MS, Shimadzu, QP 5050 A), with an MDN-5 column and a quadrupole mass spectrometer as the detector, following an injection of 2 μL. The carrier gas was helium, at a flow rate of 0.9 mL/min. The sample was measured in a split mode procedure (1:35). For GS-MS detection, an electron ionization system was used with ionization energy at 70 eV.

### 2.4. CAPeo Administration

The CAPeo mixture, in the form of two 0.5 mL soft capsules, in a concentration of 15 mL/L, was administered per os daily for two weeks (14 days) to the subjects who were allocated in the IG, while the subjects in the CG received usual care, including instructions for monitoring the progress of the illness and for taking analgesics if needed.

### 2.5. Collected Data

Data were collected on Day 1, Day 7, and Day 14. Following the initial face-to-face consultation, data collection and consultations were performed by trained medical personnel either remotely (by phone) or via home visits. The severity of symptoms was assessed through the utilization of a five-point Likert scale, starting from 0 (no severity), 1 (very mild), 2 (mild), 3 (moderate), and 4 (severe). Data were recorded on Day 1 and taken as the baseline. The primary outcome defined to assess the clinical effectiveness of the CAPeo was symptom reduction (in terms of severity and frequency), defined as the total number of symptoms over the 14 days, with measurement on Day 7 and Day 14. For reporting (and due to the small interval before the confirmed diagnosis and study inclusion), the date of the confirmed diagnosis, i.e., Day 1 of the study, was considered Day 1 for symptoms and severity and frequency assessment.

### 2.6. Statistical Analysis

Data were analyzed using the SPSS software (IBM SPSS Statistics for Windows, 2021, version 28.0 Armonk, NY, USA: IBM Corp). Frequencies of descriptive characteristics in the two groups (IG and CG) were assessed, along with the presence and severity of the symptoms. Severity was estimated as a cumulative score (summing up all symptoms’ scoring severity). Repeated-measures analysis of variance was used to assess the Δ-changes between the three days (Day 1 or baseline, Day 7, and Day 14), while the analysis of covariance was used to test the differences between the groups; the covariates used were age, gender, smoking habit, morbidity, vaccination for SARS-CoV-2, phases of the study, and use of painkillers and/or antibiotics. A critical value of 0.05 was taken as the threshold of statistical significance.

### 2.7. Ethics

In a separate file (along with authors’ details).

## 3. Results

### 3.1. Phytochemical Constituents of CAPeo

The mixture of essential oils contained carvacrol (53%), eucalyptol (13%), and β-caryophyllene (3%). Concentrations of the compounds p-cymene, γ-terpinene, borneol, and α-terpineol were 1.32, 1.17, 1.68, and 1.06%, respectively, while the concentrations of the remaining 15 compounds were less than 1%. For the complete compound analysis, please also refer to previous reports [21,22]. These concentrations refer to the stock essential oil mixture, while a concentration of 1.5% in extra-virgin olive oil (for human studies) or DMSO (Sigma-Aldrich, for cell studies) was used. This refers to the dilution of 1:1, mimicking the suggested daily dose of the CAPeo extract in humans (1 mL of 1.5% of CAPeo in olive oil) for managing upper respiratory tract infections [20,21]. As the pharmacokinetics and bioavailability of CAPeo are under investigation, we used bibliography data, suggesting a variable absorption of phenolic compounds ranging from 27 to 0.0006% and blood recovery ≤1% for the majority of compounds [27,28]. Therefore, to mimic available concentrations in humans, different dilutions (1:10, 1:100, and 1:1000 of the clinically administered concentration (15 mL extract/L, 1 mL/day) in DMSO) were used for cell studies. The same concentrations were used in a previous study to determine the protective and therapeutic effects of CAPeo in cells infected with other upper respiratory viruses [22].

### 3.2. Demographic Data and Data Regarding Health Habits

The study involved 69 patients who met the selection criteria (39 in the IG and 31 in the CG) and who completed the study (Table 1, Figure 1, Appendix A). Cycle threshold (Ct) values of all positive RT-PCR samples ranged between 11 and 26; however, no statistically significant difference was observed between IG and CG. Infection by SARS-CoV-2 virus was confirmed by rt PCR. Fifty-two percent (52%) and fifty-five percent (55%) of the IG and CG participants were women, respectively, with a mean age of 38.4 for the IG and 42.1 years for the CG. Smoking was reported in 34.2% and 29.0% of the IG and CG participants, respectively, with 36.8% of IG participants and 29.0% of CG participants reporting having at least one chronic disease. Recurrence of COVID-19 was reported for the second time in one CG patient, while 50.0% of IG (all including patients in the first phase) and 35.5% of CG reported being unvaccinated. Finally, during the illness, painkillers were used by 42.1% of IG participants and by 61.3% of CG participants. No statistically significant differences were present among the three study groups (IG_2020_, IG_2021/22_, and CG_2021/22_), with only one exception regarding the use of painkillers (Appendix A).

### 3.3. Effect on the Frequency of Symptoms

Figure 2 and Figure 3 show, in hierarchical frequency, the presence (%) of the twelve (12) local and nine (9) general symptoms, respectively, in the two groups of the study. At the onset (Day 1 or baseline), headache reporting indicated a high prevalence in both groups (IG 24 (63.2%) and CG 619 (1.3%)), decreasing in both groups on Day 7 (IG 2(5.3%) and CG 9(29.0%)) (Figure 3). The impact of CAPeo was less evident regarding the local symptoms, such as dry cough, anosmia/loss of smell, or ageusia/lack of taste on Day 14 in both groups; however, their frequency was lower in the IG (Figure 4).

### 3.4. Effect on the Severity of Symptoms

Table 2 shows the Δ-changes of the cumulative presence (number) and severity of the symptoms. A decrease in the average presence (number) of symptoms in IG (4.7 to 1.4 symptoms) as well as in CG (4.0 to 3.1) was observed on Day 7 compared with Day 1 (baseline). It represented a significantly more significant decrease in the cumulative presence (number) in IC (−3.3 vs. −0.9, *p* < 0.001; η^2^ = 0.20) on Day 7 than on Day 14 as well (−4.2 vs. −2.9, *p* = 0.027; η^2^ = 0.08). Concerning the intensity (severity) of the symptoms, there was a difference in the baseline between the IG and the CG, as shown in Table 2, with 11.58 (0.92) marginal mean in the IG in comparison with 9.16 (1.03) in the CG. Furthermore, a decrease was found in the average intensity (severity) of symptoms in both IG (11.58 to 3.44) and in CG (9.16 to 8.10), but with a significant reduction in intensity in the IG (−8.14 vs. −1.06, *p* < 0.001; η ^2^= 0.20) on Day 14 (−10.67 vs. −6.57, *p* = 0.007; η^2^ = 0.12). The observed differences remain among the three study groups (IG_2020_, IG_2021/22_, and CG_2021/22_; Appendix A) due to the finding that the two IGs recorded a similar reduction from Day 1 to Day 7 (−8.0 and −7.8 vs. −1.4, *p* = 0.006; η^2^ = 0.12) and Day 14 (−9.3 and −10.9 vs. −6.9, *p* = 0.033; η^2^ = 0.07) regarding the severity in the symptoms in comparison with the CG. To this effect, and in terms of their percentage changes (Figure 4), on Day 7 compared with Day 1 (baseline) there was a decrease in the average presence (number) of symptoms in IG by −70.2% compared with −22.5% in CG or −70.3% and 11.6%, respectively, in the severity of the symptoms.

## 4. Discussion

### 4.1. A Focus on the Main Findings and Questions Raised

This observational study revealed that general symptoms, including general fatigue, fever, and myalgia, were among the most frequent (almost fifty percent and more) reported symptoms. Regarding the local symptoms, the headache was the predominantly reported symptom. It was evident that CAPeo effectively and significantly decreased the most frequently reported general symptoms (general fatigue, weakness, fever, and myalgia) and local (headache); this difference was statistically significant on Day 7. The changes in the cumulative presence (number) and severity of symptoms also supported the positive effect of CAPeo in the patients with SARS-CoV-2.

Thus, we report that treatment with CAPeo results in an almost complete resolution of headache, fatigue, fever, and myalgia symptoms in COVID-19. Furthermore, the critical impact of CAPeo on the number and severity of symptoms is visible within seven days of initiating CAPeo administration. We, therefore, conclude that CAPeo alleviates general symptoms very quickly, with markedly better outcomes than those observed in the untreated population.

It is well known that the symptoms of a mild infection of SARS-CoV-2 gradually decrease with time; moreover, there are cases with very mild symptoms, especially with the Omicron variant. Thus, the key research question that this study needed to address is to what extent this natural product could change the evaluation of this illness and reduce the duration and severity of the symptoms.

Unfortunately, there are few studies with comprehensive reporting regarding the evolution of COVID-19 symptoms for comparison [29,30,31]. Fourteen days after COVID-19 testing, several symptoms persist, including headache, fatigue, myalgias, anosmia, ageusia, and respiratory distress.

Our study also complements a recently published review article reporting that several medicinal plants with antiviral activity may be utilized to treat viral infections or be utilized as supportive therapy [32]. Even though this review sets a limitation due to the lack of information on the safety profile and dosage of herbal treatment for various diseases, this is not the case for the CAPeo mixture used in this study (due to the evidence generated for CAPeo’s safety profile and dosage through previous studies in humans [20,21]).

To what extent does CAPeo contribute to a decrease in the probability of being infected? This is an interesting question; to that direction, we decided to include family members (father, mother, and sister of one patient) and individuals working together or living in the same house. We consider this a vital reporting aspect in establishing guidelines for sequential testing, managing oligo- or asymptomatic patients in outpatient settings, and informing future clinical study design across settings, as highlighted by a recent report [33]. The median incubation period of five days (even of approximately four and three days for the Delta and Omicron variants, respectively) creates a false sense of safety but also presents a challenge in terms of study inclusion and proper trial conduct and reporting [34]. Further research could provide key insights regarding these aspects.

This study also raises further questions regarding the antiviral properties exhibited by CAPeo. CAPeo contains 25 different microconstituents (please refer to Supplemental Table 4 of Ref. [21] for a detailed presentation of concentrations of specific constituents). The main compounds are carvacrol (53%), eucalyptol (13%), β-caryophyllene (3%), p-cymene (1.32%), γ-terpinene (1.17%), borneol (1.68%), and α-terpineol (1.06%). Recent work by our group has identified specific viral targets for CAPeo, with a direct impact on viral replication [19,22,35]. The underlying mechanism of action for this beneficial effect of CAPeo in the evolution of COVID-19 requires further in-depth study. However, at least one compound of our preparation (p-cymene) has been recently found to possess an extreme antiviral action against SARS-CoV-2 through further investigation conducted and reported by our group [35]. In addition to the possible direct antiviral effect of CAPeo, reported in cells, its beneficial effect may, at least partially, be due to its anti-inflammatory effect in humans [21]. Anosmia/loss of smell and ageusia/loss of taste, however, persisted at 13.2% in the IG at the end of the two weeks of CAPeo administration. Whether this was due to a late recovery of nasal and buccal mucosa or the persistence of the virus in a small percentage of patients [36] remains unclear. However, it is important to note that this symptom was persistent (~50%) in ambulatory patients with two negative tests in a previous study [37].

### 4.2. Limitations and Strengths

We conducted an observational study in a real-world primary healthcare setting where only patients with mild symptoms were included; a control group was also observed in this study. Certainly, our findings lack the robustness of the evidence generated by a randomized control study (RCT) and ought to be interpreted with caution. In addition, our study included patients from the last three waves of the COVID-19 pandemic in the IG. In contrast, patients from the last two waves (Delta and Omicron variants) were included in the CG. Generally, mixing patients from various waves without an equal allocation of them in both groups may introduce a selection bias that impacts the results’ interpretation. However, the secondary analysis supported the main findings of this study. Another issue that deserves further discussion is that patients not vaccinated for SARS-CoV-2 had their symptoms alleviated by the CAPeo treatment (50% in 2021/22 and all patients in 2020). This aspect may also be the reason (in terms of the contribution of the higher number and severity) for the baseline in the IG. However, the alleviation effect caused by vaccination for those not vaccinated cannot be excluded. Moreover, real-world observational data are not intended to replace RCTs; such evidence may offer valuable evidence in real-world settings while a pandemic is still unfolding. In addition, no PCR test was performed after CAPeo administration; this evidence was not included in the endpoints of study. However, we further emphasize once again that it was a PoC study and that its clinical findings are supported by laboratory evidence published elsewhere [19].

## 5. Conclusions

In conclusion, our findings from the performed clinical study, jointly with the reported in vitro studies, suggest that CAPeo (a mixture of essential oils of three Cretan aromatic plants) possesses a potent antiviral activity against SARS-CoV-2, in addition to its effect against influenza A and B, and human rhinovirus HRV14 [22]. CAPeo has an early and influential impact in alleviating the acute symptoms of COVID-19 patients. This suggests that this mixture of essential oils can be used as a complementary natural agent for patients with mild COVID-19 in primary healthcare settings. If these results are confirmed in an already-planned prospective clinical study, CAPeo may represent an effective option as a novel inexpensive therapeutic agent in cases of ambulatory patients with mild COVID-19.

## 6. Patents

SAP, CL, and EC are inventors in patents CN102762218, EP2482831, and WO2011045557, with priority numbers WO2010GB01836 20100929 and GB20090017086 20090929, related to the antiviral activity of the CAPeo.

## Figures and Tables

**Figure 1 diseases-11-00105-f001:**
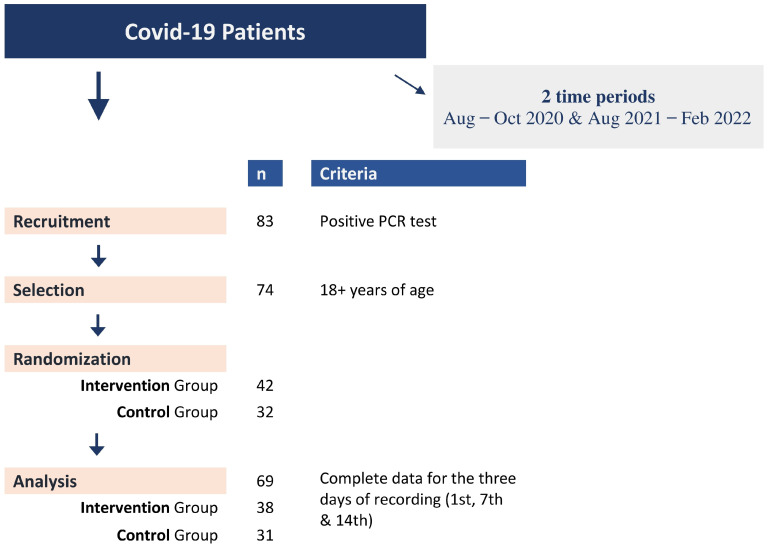
Recruitment flowchart of COVID-19 patients and randomization in intervention (IG) and control (CG) groups.

**Figure 2 diseases-11-00105-f002:**
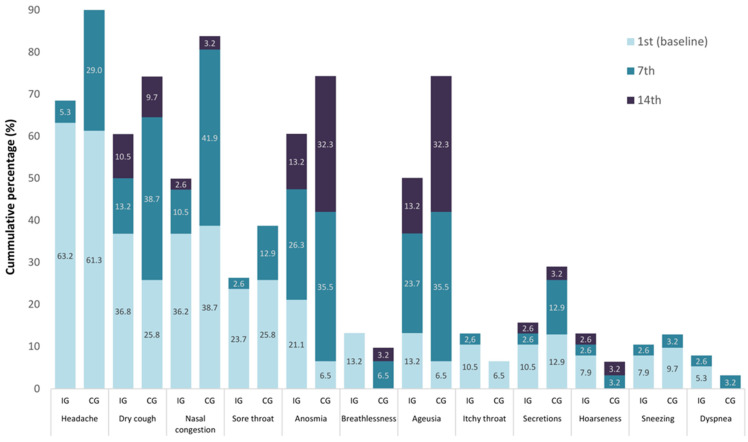
Hierarchical frequency of the presence (%) of the 12 local symptoms in intervention (IG) and control (CG) groups.

**Figure 3 diseases-11-00105-f003:**
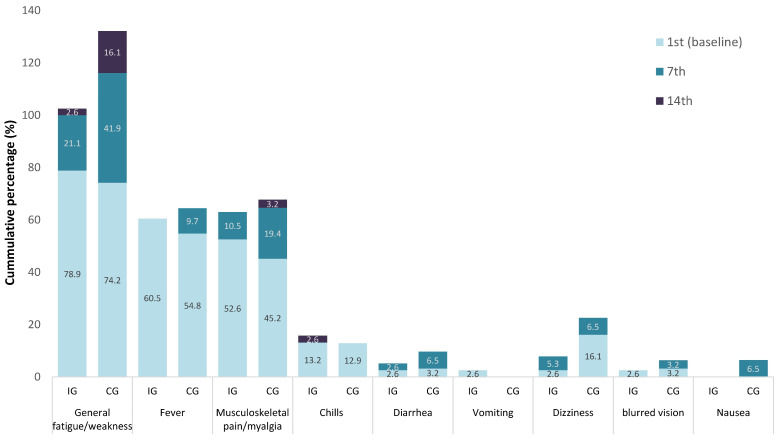
Hierarchical frequency of the presence (%) of the 9 general symptoms in intervention (IG) and control (CG) groups.

**Figure 4 diseases-11-00105-f004:**
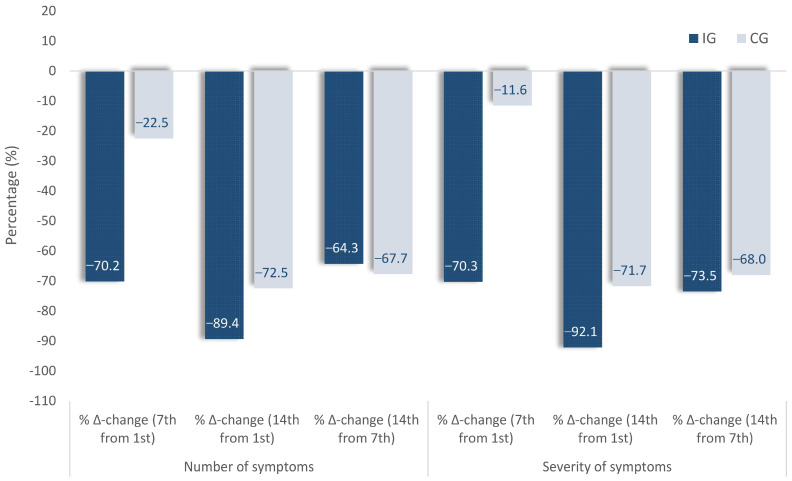
Percentage changes in number and severity of symptoms on the 7th day compared with the 1st (baseline) in intervention (IG) and control (CG) groups.

**Table 1 diseases-11-00105-t001:** Descriptive characteristics of the 69 patients in intervention and control groups.

		Groups	
		Intervention (n = 38)	Control (n = 31)
		n (%)	
Gender	males	18 (47.4)	14 (45.2)
	females	20 (52.6)	17 (54.8)
Age, years	mean age ± stand.dev.	38.4 ± 12.8	42.1 ± 16.1
Family members	yes	11 (28.9)	15 (48.4)
Study phases	August–October 2020	13 (34.2)	-
	August 2021–February 2022	25 (65.8)	31 (100.0)
Smokers	non	22 (57.9)	13 (42.0)
	former	3 (7.9)	9 (29.0)
	current	13 (34.2)	9 (29.0)
Morbidity (at least one chronic disease)	yes ^a^	14 (36.8)	9 (29.0)
Recurrence of COVID-19 (in last six months)	yes	--	1 (3.2)
Vaccination for SARS-CoV-2 (doses)	none	19 (50.0)	11 (35.5)
	one	5 (13.2)	3 (9.7)
	two	10 (26.3)	14 (45.2)
	three	4 (10.5)	3 (9.7)
Administration/intake of medicinal or other compound for the symptoms (before, at the point of, or following inclusion to the study, additionally to CAPeo)	painkillers	16 (42.1)	19 (61.3)
	antibiotics	4 (10.5)	--

^a^ Includes nineteen different groups of diseases, e.g., hypertension, diabetes mellitus, heart disease, cancer, etc.

**Table 2 diseases-11-00105-t002:** Mean changes in number and severity of symptoms between intervention and control groups on Day 7 and Day 14 in relation to baseline (Day 1).

		Groups			
		Intervention	Control		
Symptoms	Days of Follow-Up	Marginal Means (Stand. Errors)		*p*-Value	η^2^
Number	1st (baseline)	4.7 (0.3)	4.0 (0.4)		
	7th	1.4 (0.4)	3.1 (0.4)		
	Δ-change (7th from 1st)	−3.3 (0.4)	−0.9 (0.5)	<0.001	0.20
	14th	0.5 (0.2)	1.1 (0.3)		
	Δ-change (14th from 7th)	−0.9 (0.3)	−2.1 (0.3)	0.008	0.16
	Δ-change (14th from 1st)	−4.2 (0.3)	−2.9 (0.4)	0.027	0.08
Severity	1st (baseline)	11.58 (0.92)	9.16 (1.03)		
	7th	3.44 (1.07)	8.10 (1.20)		
	Δ-change (7th from 1st)	−8.14 (1.14)	−1.06 (1.28)	<0.001	0.20
	14th	0.91 (0.55)	2.59 (0.62)		
	Δ-change (14th from 7th)	−2.53 (0.73)	−5.51 (0.82)	0.014	0.10
	Δ-change (14th from 1st)	−10.67 (0.91)	−6.57 (1.02)	0.007	0.12

Based on 12 topics and 9 general symptoms. The severity based on a five-point Likert scale as: 0 = none, 1 = very mild, 2 = mild, 3 = moderate, and 4 = severe. Scores of mean changes were extracted as summing up all symptoms’ intensity. Comparisons of Δ-changes between groups were performed by using repeated measures analysis of covariance and the covariates used were age, gender, smoking habit, morbidity, vaccination for SARS-CoV-2, phases of the study, and use of analgesics or and antibiotics.

## Data Availability

Data available on request due to restrictions eg privacy, or ethical reasons.

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
