# Peer review of "A Mixture of Essential Oils from Three Cretan Aromatic Plants Inhibits SARS-CoV-2 Proliferation: A Proof-of-Concept Intervention Study in Ambulatory Patients"

_diseases, 2023, doi:10.3390/diseases11030105_

Round 1
Reviewer 1 Report
This observational study of the herbal essential oil (CAPeo) from three herbal combinations in alleviating the general and local symptoms of COVID-19 patients. This Proof-of-Concept study was supported by the previous studies of the phytochemical analysis, safety and the potential antiviral functions on influenza and SARS-CoV-2-infected cells. The CAPeo could be developed as the health supplement and/or as drug after prospective clinical study. There are numbers of studies reported the antiviral functions of essential or volatile oils on SARS-CoV-2, however the studies of extracts or phytochemicals from the three herbal plants against or preclinical application in COVID-19 is limited. This study is original and valuable for publication.
There some concerns are required for clarification before publication (see below).
Major comments:
1. Authors mentioned that SARS-CoV-2 infection was confirmed by a real-time PCR test. If the Ct value in qPCR was available, it is better to be included in the Demographic data.
2. It seems that there was no PCR test performed after CAPeo administration or in the endpoint of study. This data is valuable in supporting the antiviral function of CAPeo in COVID-19 patients.
3. Since the study was not involved the placebo group as the control group (e.g., soft gels with olive oil only), the bias on the intervention group should be discussed, may be in the “limitation” part, in particular data collected by face-to-face or phone consultations for general symptoms.
4. It is surprising that non-vaccinated patient for SARS-CoV-2 could were alleviated by the CAPeo treatment (50% in 2021/22 and all patients in 2020). It is suggested to discuss the alleviation effect caused by vaccination. It might be the reason in the contribution of the higher number and severity of the baseline in the intervention group.
5. The discussion parts are rather discrete. Discussion section combines with Results section can be considered, or reorganized into specific subtitles.
6. In the discussion (line 273), the statement “No side effects have been reported” might be overstated. There was no control group which healthy people used or treated with CAPeo in their previous studies.
Minor comments:
1. Line 61: “In vitro” should be italic.
2. Line 71: typo for “Proof-of-of-Concept (PoC)”. It should be “Proof-of-Concept (PoC)”.
3. The typo of “Control (IG)” groups. It should be “Control (CG) groups” in the Figure 1 (Line103), Figure 2 (Line180), Figure 3 (Line 204), Figure 4 (Line 208).
4. Line165-166: The statement “…..while their chemical structure is presented in Figure 2.”. Figure 2 is showing the hierarchical frequency of the presence of the local symptoms. Or “Figure 2” was from the previous reports?
5. Line 200: typo for “CaPeo”. It should be “CAPeo”.
6. Line 214: It is suggested to rephrase the sentence. “It represents a significantly more significant decrease in the cumulative presence (number) in IC”.
Nil.
Author Response
Dear Reviewer,
We appreciate your kind comments and suggestions. They have been considered and our point-by-point responses to your comments appear in the attached file.

Reviewer 2 Report
This is a good manuscript that includes interesting information about one of the most challenging and important dares for Medicine in our era. However, the manuscript includes several problems and requires improvements. So, I recommend accepting this article after REVISIONS.
1. In lines 40-42, “Since the COVID-19 pandemic outbreak in 2019, a previously unseen international effort has been undertaken to identify the underlying cause (the SARS-CoV-2 virus) and for the detailed analysis of its genome [1].” please change with “Since the coronavirus disease 2019 [COVID-19] pandemic outbreak in 2019, a previously unseen international effort has been undertaken to identify the underlying cause (the severe acute respiratory syndrome coronavirus 2 [SARS-CoV-2]) and for the detailed analysis of its genome [1].”.
2. The manuscript includes several mistakes from a template point of view. For example, References section uses a different format than indicate in the template document for “diseases”. The authors have to change the format and use the indicated in the template.
3. In lines 42-43, “A global effort is directed toward efficient therapy [2-4] or the development of efficient vaccines [5, 6].” Please cite relevant literatures published in 2023. (For example, DOI: 10.1038/s41467-023-40018-1; DOI: 10.1016/j.ejmech.2023.115503; DOI: 10.1136/bmj.p1111).
4. In lines 45-48, Among the many products tested against COVID-19, several natural products (DOI: 10.3390/ijms24119589; DOI: 10.3389/fphar.2022.926507), including herbal extracts (DOI: 10.1016/j.jep.2021.113869; DOI: 10.1016/j.jff.2023.105544), have also been assayed (critically reviewed in [10] and references therein), targeting mainly the viral proteases.
5. In “Discussion” section, the quality must be improved.
6. “Figures” should be improved.
Minor editing of English language required
Author Response
Dear Reviewer,
We appreciate very much your comments and suggestions. All have been considered in the revised manuscript. Our point-by-point responses to your comments appear in the attached file.
